# Preparation of Ag@ZIF-8@PP Melt-Blown Nonwoven Fabrics: Air Filter Efficacy and Antibacterial Effect

**DOI:** 10.3390/polym13213773

**Published:** 2021-10-31

**Authors:** Bing-Chiuan Shiu, Ying Zhang, Qianyu Yuan, Jia-Horng Lin, Ching-Wen Lou, Yonggui Li

**Affiliations:** 1Fujian Key Laboratory of Novel Functional Fibers and Materials, Minjiang University, Fuzhou 350108, China; bcshiu@mju.edu.cn; 2Fujian Engineering Research Center of New Chinese Lacquer Material, College of Material and Chemical Engineering, Minjiang University, Fuzhou 350108, China; 3Innovation Platform of Intelligent and Energy-Saving Textiles, School of Textile Science and Engineering, Tiangong University, Tianjin 300387, China; zhangying8384@126.com (Y.Z.); yuan_qianyu@163.com (Q.Y.); 4Advanced Medical Care and Protection Technology Research Center, College of Textile and Clothing, Qingdao University, Qingdao 266071, China; 5Laboratory of Fiber Application and Manufacturing, Department of Fiber and Composite Materials, Feng Chia University, Taichung 40724, Taiwan; 6School of Chinese Medicine, China Medical University, Taichung 40402, Taiwan; 7Department of Fashion Design, Asia University, Taichung 41354, Taiwan

**Keywords:** flame-retardant nonwoven fabric, metal–organic framework, ZIF-8, antibacterial nonwoven fabric, air filter nonwoven fabric

## Abstract

Serving as matrices, polypropylene (PP) melt-blown nonwoven fabrics with 4% electrostatic electret masterbatch were incorporated with a 6%, 10%, 14%, or 18% phosphorus-nitrogen flame retardant. The test results indicate that the incorporation of the 6% flame retardant prevented PP melt-blown nonwoven fabrics from generating a molten drop, which, in turn, hampers the secondary flame source while increasing the fiber diameter ratio. With a combination of 4% electrostatic electret masterbatch and the 6% flame retardant, PP melt-blown nonwoven fabrics were grafted with ZIF-8 and Ag@ZIF-8. The antibacterial effect of ZIF-8 and Ag@ZIF-8 was 40% and 85%, respectively. Moreover, four reinforcing measures were used to provide Ag@ZIF-8 PP melt-blown nonwoven fabrics with synergistic effects, involving lamination, electrostatic electret, and Ag@ZIF-8 grafting, as well as a larger diameter because of the addition of phosphorus-nitrogen flame retardants. As specified in the GB2626-2019 and JIS T8151-2018 respiratory resistance test standards, with a constant 60 Pa, Ag@ZIF-8 PP melt-blown nonwoven membranes were tested for a filter effect against PM 0.3. When the number of lamination layers was five, the filter effect was 88 ± 2.2%, and the respiratory resistance was 51 ± 3.6 Pa.

## 1. Introduction

Melt-blowing nonwoven fabrics is an advanced production technique used to create ultrafine fibers out of a polymer melts, drawn from the mold of a screw extruder, the polymer undergoes high-speed hot air jet stretching, and then the ultrafine fibers are entangled and aligned randomly in a turbulent airflow, forming nonwoven fabrics that are suitable for diverse applications, including air filtration [1], oil–water separation [2], tissue engineering [3], heavy metal adsorption [4], sound insulation materials [5], and battery separators [6]. Following the spread of COVID-19, airborne viruses will permanently contaminate human society and so will other pollutants, e.g., PM 2.5. To address this issue, melt-blown nonwoven fabrics that consist of ultrafine fibers in a randomly entangled structure ensure an excellent air filtration effect. The four mechanisms of physical filters to capture aerosol particles are gravity settling, inertia impact, interception, and Brownian diffusion, all of which exert a crucial effect on aerosol particles in a specified size range [7,8].

Being material-based fibrous filters, electret filters feature Semipermanence [9] and are commonly used as a medium for air filters in the market, thereby satisfying the high demands of collection efficiency. The particle size at the lowest filtration efficiency, named the most penetrating particle size (MPPS), typically around 0.3 µm or smaller, is used to determine the dominant capture mechanism of an air filter [10]. Electret filters, with quasi-permanent electric charges on the fibers and additional electrostatic attraction, show a higher initial filtration efficiency and a much lower pressure drop compared to mechanical high-efficiency particulate air (HEPA) filters. They have been widely applied in indoor air cleaners and heating, ventilating, and air conditioning (HVAC) systems for ensuring a high quality of indoor air [11]. Several experiments have been carried out to explore the factors that cause the efficiency degradation of electret filters. The filtration efficiency of an electret fiber can be approximated as the sum of the electrostatic and mechanical collection efficiencies, where a superior electret melt-blown filtration material has a high filtration efficiency of 99.65%. The electret time, electret voltage, and electret distance are three important process parameters that affect the electret effect. As the electret time increases, the equivalent surface charge density of the deposition increases, and the potential on the electret surface rises. After the electret time increases again, when the surface potential of the filter material is high enough, the charge under the needle tip will be repelled to move to other places with a lower charge density [12]. When the electret ends, the charge surface density reaches the saturation state, and thus when the electret time increases again, the filtration efficiency of the filter material does not change significantly. Electrostatic electret equipment, dust, bacteria, and viruses in the air are attached to the particles, which are mainly negatively charged. The melt-blown cloth is positively charged, and it is easy to adsorb these negatively charged particles. Melt-blown cloth electrostatic electret equipment is a special type of equipment that generates static electricity. It offers a stable output voltage, comprehensive protection, simple operation, high efficiency, low flow resistance, antibacterial properties, and energy saving. It guarantees the physical collision blocking effect of conventional filter materials and increased electrostatic adsorption.

On the other hand, metal–organic frameworks (MOFs) are newly emerging materials that are characterized by a high porosity, a high specific surface area, a porous crystal area, and a highly adjustable aperture [13]. Due to their great potential, MOFs have garnered much attention in the fields of air storage, separation, catalysis, electronics, sensors, and medicine [14,15]. In particular, zeolitic imidazolate framework-8 (ZIF-8) is one major MOF that presents unique properties, such as high heat resistance, chemical stability, a high surface area, permanent porosity, and a high adsorption capacity [16], the attributes of which make ZIF-8 a good candidate for industrial gas separation, storage, and catalytic uses [17]. The in-situ growth method helps load ZIF-8 efficiently while maintaining its pristine structure/activity, which, in turn, contributes to its valuable industrial applications [18]. Hence, previous studies employed the in-situ growth method to combine Ag-MOFs@CNF@ZIF-8 with biodegradable cellulose-based filters. The materials demonstrated reinforcement in the filter efficiency against PM2.5 from 44% to 94.30% as well as a pressure drop from 19 to 158 Pa [19]. In another study, ZIF-8@CF filters, a composite type, exhibited a filtration efficiency of 44% for 0.3 μm particles and 65% for 0.5 μm particles, that is, the filtration efficiency was improved by 98.36% and 99.94% with the pressure increasing from 21 to 134 Pa [20]. PPC composite melt-blown fibrous membranes loaded with ZIF-8 nanocrystals were prepared by the in-situ polymerization method under mild conditions. The composite membranes demonstrated a high PM 2.5 filtration efficiency (PPC/ZIF8-9 membrane, 91.68 ± 0.57%) as well as a low pressure drop (PPC/ZIF8-9 membrane, 45 Pa). Notably, the PM 2.5 filtration efficiency of the composite membranes was enhanced by nearly 32.83% compared to that of the pure PPC filter, but the pressure drop was not increased [21].

Polypropylene (PP) melt-blown nonwoven fabrics are popular, serving as the filter layer for diverse facial masks. PP melt-blown nonwoven fabrics possess extraordinary dust filtration performance that protects the skin. They are also used as a protective layer in medical protective clothing and protective apparel because of their excellent chemical resistance and breathability [22]. Nowadays, due to the outbreak of COVID-19, medical protective clothing is required to block body fluids, blood, secretions, particulate matter, and aerosols and possess a powerful antibacterial effect, too. Therefore, in this study, 4% electrostatic electret masterbatch was employed to provide PP melt-blown nonwoven membranes with a strengthened electret effect, while phosphorus-nitrogen flame retardants (6%, 10%, 14%, and 18%) were used to improve the low combustion resistance of PP melt-blown nonwoven membranes. Next, ZIF-8/Ag was grafted with the filter group containing the 6% flame retardant via a solvent method, thereby strengthening the filtration efficacy against PM 0.3, and the antibacterial effect. In addition, the resulting filters were evaluated for their air filtration effect and respiratory resistance rate as related to the number of lamination layers. From the results, Ag@ZIF-8@PP melt-blown nonwoven fabrics can be considered for use in both air filters and the protective layer of medical protection gear. 

## 2. Experimental

### 2.1. Material

The polypropylene (PP) resin (Dongguan Xiangsheng Plastic Co., Ltd., Guangdong, China) had a melt flow rate (MFR) of 35 g·10 min^−^^1^. PP flame-retardant masterbatch (RSPP-100M, bromine content: 16%, nitrogen content: 6%, phosphorus content: 4%) and PP electrostatic electret masterbatch were purchased from Rise Chemical Technology Co., Ltd., Shanghai, China. Zinc nitrate hexahydrate (AR, Zn (NO_3_)_2_ × 6H_2_O) and methanol (AR, CH_3_OH) were purchased from Kerris (Fine Chemical Co., Ltd., Tianjin, China). 2-Methylimidazole (98%, C_4_H_6_N_2_) was purchased from (Sigma Aldrich, Saint Louis, MO, USA). All chemical reagents were used without further purification. The nanosilver antibacterial agent was purchased from Xing Zhou Chemicals Co., (Ltd., Shanghai, China).

### 2.2. Preparation of Melt-Blown Nonwoven Fabrics and Electrostatic Electret

Figure 1a shows melt-blown nonwoven fabrics that were prepared by a single-screw melt blowing apparatus (SJ45X36, Keshengda Plastic Machinery Co., Ltd., Qingdao, China). The processing parameters are listed in Table 1. The as-prepared masterbatch was named PP, PF-6, PF-10, PF-14, PF-18, and PF-22 according to the content of flame retardant in the masterbatch. As suggested by the supplier of electrostatic electret masterbatch, electrostatic electret masterbatch has a specified content of 4%. Each batch of samples was prepared as follows: PP pellets (3 kg) that were blended with the flame-retardant masterbatch and electrostatic electret masterbatch in advance were fed into the barrel of the machine. Figure 1b shows the electrostatic electret machine (ESD-RPB, Lishan Technology Co., Ltd., Shanghai, China) equipped with a voltage of 30 kv.

### 2.3. Preparation of MOF and Ag@MOF

As shown in Figure 2, zinc nitrate hexahydrate (3 mmol) and 2-methylimidazole (15 mmol) were separately combined with anhydrous methanol, after which the two blends were mixed for 3 h and kept for 8 h. Next, the final mixture was processed with centrifugation at 5000 rpm for five minutes, and then rinsed with methanol three times. After, the supernatant was removed, and the mixture was dried in a vacuum drying oven at 80 °C, thereby obtaining white MOF powders.

Zeolitic imidazolate framework-8 (ZIF-8) is one major MOF type, and Ag@MOF was prepared as follows. The nanosilver (Ag) antibacterial agent was added to the final mixture (produced as described in the previous paragraph) with 3 h of mixing and 8 h of storage, after which the Ag-containing blend underwent centrifugation, rinsing, and drying, thereby yielding Ag@ZIF-8 [7,21,23].

### 2.4. Preparation of ZIF-8@ and Ag@ZIF-8@ Melt-Blown Nonwoven Fabrics

ZIF-8 and Ag@ZIF-8, weighing 40% of the nonwoven fabric, were separately added to anhydrous methanol for dissolution, totaled 40% of the weight of the nonwoven fabric. They. Next, a PP melt-blown nonwoven fabric (10 × 10 cm) was immersed completely in a ZIF-8 mixture or an Ag@ZIF-8 mixture for 24 h, after which deionized water was used to rinse the MOF that did not bond well with the fabrics. The wet PP melt-blown nonwoven fabrics were dried at 80 °C in order to gain a constant weight, forming ZIF-8@ melt-blown nonwoven fabrics and Ag@ZIF-8@ melt-blown nonwoven fabrics.

### 2.5. Characterization

The IR spectra of the PP melt-blown nonwoven fabric membranes were detected by an FT-IR spectrometer (MPIR8400S, Shimadzu, Kyoto, Japan) based on the ATR method. Sixteen scans were conducted for each melt-blown nonwoven fabric at a resolution of 4 cm^−1^. The SEM images of the as-prepared PP melt-blown nonwoven fabrics were photographed by a field-emission SEM (Nova Nano SEM 230, FEI, Hillsboro, OR, USA) at an acceleration voltage of 2 kV. The fiber size distribution (fiber diameter) was measured with Image-Pro Plus 6.0 software. The antibacterial efficacy of ZIF-8@ nonwoven fabrics was evaluated individually using a quantitative test as specified in AATCC-100 and a qualitative test as specified in JISL1902. Five samples for each specification were used. Gram-positive *Staphylococcus aureus* (AATCC 25922TM) and Gram-negative *colibacillus* (AATCC 25922TM) were used for the tests. The Ag@ZIF-8@ nonwoven fabrics were the experimental group, while the PP nonwoven fabrics were the control group. The qualitative test used the inhibition zone to determine the antibacterial effect, and the quantitative test used Equation (1) to determine the bacteria reduction rate (BR%).
BR% = (A − B)/A(1)
where A stands for the number of colonies of the control group, and B stands for the number of colonies of the experimental group.

For the vertical combustion test, the YG(B)815D-I vertical combustion instrument produced by Darong Textile Instrument Co., Ltd. (Wenzhou, China) was used. Trimmed to 80 × 300 mm according to GB/T5455-2014 test standards, the nonwoven fabric was fixed on the test clamp, the flame height of methane was specified as 40 mm in the experiment, and the experiment started when the flame was stable. The flame was ignited under the nonwoven fabrics for 10 s and then automatically moved away, after which skimmed cotton was positioned under the sample to monitor whether droplets are produced.

The specific surface and pore size of the material were measured by an automated gas sorption analyzer (Autosorb-iQC). Before the adsorption test, the material was blown with nitrogen at 350 °C for 5 h to remove the vapor and impurities. The air filtration efficiency test was conducted as specified in GB2626-2019 and JIS T8151-2018 using a mask filter material particle filtration efficiency tester (FE/R-2626-III, LiSan Technology Co., Ltd., Shenzhen, China), with the size of non-oily suspended particles being 0.3 µm. All experiment data tests were conducted at least five times, and the data are presented as means ± SD. Statistical analyses were performed using GraphPad Prism (GraphPad 4 Journal of Biomaterials Applications 0 (0) Software, Inc., San Diego, CA, USA). Data were analyzed using one-way analysis of variance (ANOVA), followed by the two-tailed *t*-test for comparison between two groups. The threshold for statistical significance was *p* < 0.05.

## 3. Results and Discussion

### 3.1. SEM Analysis

Figure 3 presents SEM images showing the morphology of the samples. PP melt-blown nonwoven fabrics containing 4% electrostatic electret masterbatch showed a nanofiber diameter that is in direct proportion to the content of the phosphorus-nitrogen flame retardant. Compared to the pure PP melt-blown nonwoven fabrics in Figure 3a, the presence of the flame retardant resulted in an irregular fiber distribution in the melt-blown nonwoven fabrics, as shown in Figure 3b–e. The uneven nanofiber diameters generated a fiber morphology that is beneficial for the ventilation and the air filter efficiency [7,21,23]. The average fiber diameter for each group is PP (8.89 µm), PP-6 (11.89 µm), PP-10 (14.94 µm), PP-14 (16.03 μm), and PP-18 (17.26 µm). Serving as a filter, the optimal PP melt-blown nonwoven fabrics were those with a nanofiber diameter of 2–14 µm [24]. Regardless of whether the flame retardant was incorporated, the average thickness of PP melt-blown nonwoven fabrics was 0.46 ± 0.3 mm, with an average weight of 0.92 mg/cm^2^. Although the presence of the flame retardant did not affect the weight of the melt-blown nonwoven fabrics, the nanofibers with various thicknesses were interlaced irregularly, which caused the fabrics to have a rugged morphology and a rough texture. 

### 3.2. Vertical Burning Test

Figure 4 shows the vertical burning test results of different PP melt-blown nonwoven fabrics. It can be observed that the groups consisting of the phosphorus-nitrogen flame retardant were highly inflammable. The burning test process involved nonwoven fabrics being placed over the clamps in the combustion test equipment and the skimmed cotton being placed beneath the clamps. The flame continued to burn over the nonwoven fabrics, after which the molten drop set the skimmed cotton alight, causing the second flame source. A tremendous amount of medical-related protective clothing is now being used because of the COVID-19 pandemic. Nonwoven fabrics for disposable medical wear, medical masks, and medical protective wear do not require the flame retardant effect, but CPSIA (US), Oeko-tex100 (EU), and GB31701-2015 (China) require infant-related clothing products to be flame retardant.

The PP melt-blown nonwoven fabrics were incredibly flammable. By contrast, the PP-6 melt-blown nonwoven fabrics did not generate a molten drop in the vertical burning test. Although the damage length was not significantly improved, the incorporation of the 6% flame retardant exerted a positive influence on the phenomenon of molten drop. Due to the presence of a flame retardant, the SEM images (Figure 3) indicate that the interlaced fibers have various diameters, which, in turn, favors air filtration. Therefore, the specified PP melt-blown nonwoven fabric (i.e., PP-6) that is composed of 4% electrostatic electret masterbatch and the 6% phosphorus-nitrogen flame retardant was used in the subsequent measurements.

### 3.3. SEM Analysis of ZIF-8@ Melt-Blown Nonwoven Fabrics

Figure 5 shows the SEM images of the ZIF-8@ melt-blown nonwoven fabrics and Ag@ZIF-8@ melt-blown nonwoven fabrics, where both groups were prepared with the PP-6 group as the matrices. There are many granular cubes in Figure 5a, the magnified image of which is Figure 5b, where there are many crystal particles adhered to the melt-blown nonwoven fabrics. Furthermore, the EDS analysis confirms that the crystals exhibit the characteristic peaks of ZIF-8 [25,26,27]. ZIF-8 itself has a minor antibacterial effect [28], and it can increase the nonwoven fabric’s interception of suspended particles [21]. To compensate for the disadvantage of ZIF-8, a nanosilver antibacterial agent was incorporated to form Ag@ZIF-8@ melt-blown nonwoven fabrics. As a result, the SEM in Figure 5c proves the adhesion of the crystals to the fibers, while the EDS analysis also substantiates the presence of Ag. 

The FTIR spectra of PP-6, ZIF-8@PP-6, and Ag@ZIF-8@PP-6 are shown in Figure 6. The C–N absorption peak takes place at around 1255 cm^−1^, and the stretching vibrations of C–N and C==C of the imidazole ring occur at 1250 cm^−1^ and 1562 cm^−1^, respectively [4]. The stretching vibration or bending vibration of the imidazole ring is presented at 591–1550 cm^−1^; the stretching vibration of C=N in the imidazole ring is presented at 1580 cm^−1^; and the stretching vibration of C–H bonds for the imidazole ring in the aromatic and aliphatic series is presented at 2930–3140 cm^−1^ [7]. The characteristic peak at 1146 cm^−1^ is ascribed to the aromatic C–N stretching mode. The in-plane bending mode of the imidazole ring and the aromatic bending mode of sp^2^ C–H are assigned to the signals at 994 and 760 cm^−1^, respectively. Similarly, the peak at 684 cm^−1^ is derived from the out-of-plane bending vibratioof the 2-methylimidazole ring [25,29].

### 3.4. BET Analysis of ZIF-8@ Melt-Blown Nonwoven Fabrics

The ZIF-8 organic–metal framework material possesses a huge surface area, a high load efficiency, and a controllable porosity. These attributes prevent effective materials, e.g., antibodies, from being affected by the interference of the ambient temperature, and pressure. By means of N^2^ adsorption–desorption experiments, the pore structure parameters of ZIF-8@ melt-blown nonwoven fabrics, including the specific surface area and aperture, are characterized in Figure 7, as related to the antibacterial agent. Figure 7a–d individually show the pore structure parameters before and after loading the ZIF-8 nanomaterial. According to the adsorption–desorption isotherm in Figure 7a, with P/P0 < 0, the adsorption capacity swiftly increases and fills the microporous area of the fillers in the beginning. Afterwards, with P/P0 > 0.1, the adsorption capacity increases mildly, suggesting that the micropore filling almost approaches saturation. To sum up, the maximal specific area of ZIF-8 is as high as 1010.33 m^2^/g, which makes the ZIF-8 nanomaterial a popular adsorbent. Figure 7c shows that the ZIF-8 nanomaterials have a pore size of 5.94 nm that is classified as belonging to the mesoporous structure range (micropore < 2 nm, 2 nm < mesoporous structure < 50 nm).

Figure 7b shows the adsorption–desorption isotherm when ZIF-8 was loaded with an antibacterial agent. With P/P0 < 0.1, when the material is in contact with air, the air is first adsorbed. With P/P0 > 0.1, there is a distinct platform area that suggests that the material reaches the maximal gas adsorption capacity. Due to the presence of van der Waals force among the molecules, multi-layered air adsorption occurs and is accompanied by condensation of pores. Subsequently, the adsorbed gas enters the micropores and is then rendered with liquefaction, which, in turn, boosts the adsorption capacity. Figure 7b shows a spike in the adsorption isotherm, while ZIF-8 that was loaded with an antibacterial agent exhibits a decreased specific surface area of 666.96 m^2^/g and an increased pore diameter of 9.00 nm. In conclusion, the antibacterial agent was successfully loaded on the ZIF-8 nanomaterial, and it filled the comparatively smaller pores of ZIF-8.

### 3.5. Antibacterial Effects of Ag@ZIF-8@ Melt-blown Nonwoven Fabrics

There was a dire shortage of medical surgical masks at the beginning of the current pandemic when COVID-19 started escalating. PP melt-blown nonwoven fabrics now have a surplus in supply because of their excessive production. Therefore, this study aimed to broaden the application range of PP melt-blown nonwoven fabrics. Moreover, the flame retardant effect and antibacterial effect were incorporated so the PP melt-blown nonwoven fabrics can also be used in medical protective clothing. Ag@ZIF-8@ melt-blown nonwoven fabrics and ZIF-8@ melt-blown nonwoven fabrics are thus proposed. The antibacterial test results are shown in Figure 8, where (a1, b1) are the control group, while (a2, b2) are the ZIF-8@ melt-blown nonwoven fabrics against *Escherichia coli* and *Staphylococcus aureus*, respectively. ZIF-8 demonstrated a slight antibacterial effect [30,31] that was about 30~40%. Figure 8(a3, b3) show that the presence of the nano silver antibacterial agent strengthened the antibacterial effect significantly, which was over 85%. ZIF-8 had a limited loading capacity, which, in turn, restricted the loading of the nano-Ag antibacterial agent. In its current state, medical protective clothing is repetitively used, meaning it needs to be sterilized using alcohol or UV lights. Being electrostatic electret based, the proposed PP melt-blown nonwoven fabrics are an effective air filter that intercepts airborne aerosols efficiently [32,33]. Protective clothing with PP melt-blown nonwoven fabric as the surface layer is air permeable and can block microbe-containing aerosols. Furthermore, PP melt-blown nonwoven fabrics can be treated again with electrostatic electret to be saturated with static charges, during which bacteria and viruses are also removed simultaneously. As shown in Figure 9, a bacterial suspension with a concentration of 10^6^ was dripped over the Ag@ZIF-8@ melt-blown nonwoven fabrics, which were then processed with electrostatic electret. With a voltage of 30 kv, the elective was left for five seconds and then removed, providing Ag@ZIF-8@ melt-blown nonwoven fabrics with an antibacterial rate of <1%.

### 3.6. Filtration Efficacy of Airborne Suspended Aerosols

As specified in GB2626-2019 and JIS T8151-2018, NaCl particulates are used to simulate aerosols with a diameter of around 0.3 µm, and the test respiratory resistance should not exceed 60 Pa. The control group was the melt-blown nonwoven fabrics of commercially available 3M medical masks. All samples in Figure 10 were processed with electrostatic electret, and the PP melt-blown nonwoven fabrics only weighed 9.2 mg/m^3^, with a respiratory resistance of 2 Pa. Next, the proposed PP melt-blown nonwoven fabrics (containing 4% electrostatic electret masterbatch) were laminated. After being processed with electrostatic electret, the five-layered PP melt-blown nonwoven fabrics showed 51% air filtration against 0.3 µm aerosols. The melt-blown layer of 3M masks showed a filtration efficiency of 63% and a respiratory resistance of 40 Pa. 

After a flame retardant agent was added to PP-6, the resulting PP melt-blown nonwoven fabrics had a greater diameter ratio that improved the filtration effect, and the five-layer lamination helped the materials to gain a filtration efficiency of 70% and a respiratory resistance of 33 Pa. By contrast, five-layered Ag@ZIF-8@ melt-blown nonwoven fabrics showed a filtration efficiency of 88% and a respiratory resistance of 51 Pa. A filter effect of ≥95% is demanded by international standards. Both medical masks and medical protective clothing employ multiple laminations, combining spunbond nonwoven fabric layers or needle-punched nonwoven fabric layers, to improve the air filtration effect via the multilayer structure. In addition to the air filtration effect, composite layers can also prevent blood, bodily fluid, and secretions from contacting the human body.

## 4. Conclusions

The production capacity of melt-blown nonwoven fabrics has been boosted excessively in the wake of COVID-19, and therefore the yield surplus of melt-blown nonwoven fabrics now demands to be more effectively and properly applied. In this study, 4% electrostatic electret masterbatch was used to strengthen the electret function, after which a phosphorus-nitrogen flame retardant was added to improve the flame retardant effect, increasing the fiber diameter ratio that strengthens the air filtration. Based on the content of the flame retardant, PP-6, PP-10, PP-14, and PP-18 melt-blown nonwoven fabrics demonstrated comparable vertical combustion resistances, which determined PP-6 as the specified group for subsequent comparisons. PP melt-blown nonwoven fabrics did not generate a molten drop or a secondary flame source, qualifying their application in medical protective clothing.

Using the solvent method, Ag@ZIF-8@ melt-blown nonwoven fabrics demonstrated an antibacterial effect of 80–85%. Following the development of electrostatic electret machines, it is becoming more convenient to process electrostatic electret, and therefore this study proposes PP melt-blown nonwoven fabrics that can be made into medical protective clothing. For sterilization and repetitive use, electrostatic electret treatment can be conducted to fill the PP melt-blown nonwoven fabrics with static charges. During this process, the bacteria and microbes can be killed effectively with the PP melt-blown nonwoven fabrics being loaded with static charges, which sufficiently protects medical staff and thoroughly afterwards.

The respiratory resistance of a single-layer PP melt-blown nonwoven fabric was as low as 2–5 Pa, and electrostatic electret contributed only 40–45% to air filtration. However, the incorporation of a phosphorus-nitrogen flame retardant increased the fiber diameter ratio; Ag@ZIF-8 can be grafted over PP-6 melt-blown nonwoven fabrics that can then be laminated with multiple layers, thereby strengthening the air filtration. The proposed PP melt-blown nonwoven fabrics with a low weight per unit area contribute to a great air filtration effect and manageable respiratory resistance. When used in medical protective clothing or medical masks, if other layers have good air filtration, a high respiratory resistance, and a low air permeability, only the number of layers of PP melt-blown nonwoven fabrics needs to be reduced. Conversely, if other composite layers emphasize air permeability and other functions but the air filtration is low, the number of layers of Ag@ZIF-8@ melt-blown nonwoven fabrics needs to be increased to achieve the best compounding effect. The test results indicate that five-layered Ag@ZIF-8@ melt-blown nonwoven fabrics attained an air filtration value of 85–88%, while the four-layered fabrics attained an air filtration value of 70–74%.

## Figures and Tables

**Figure 1 polymers-13-03773-f001:**
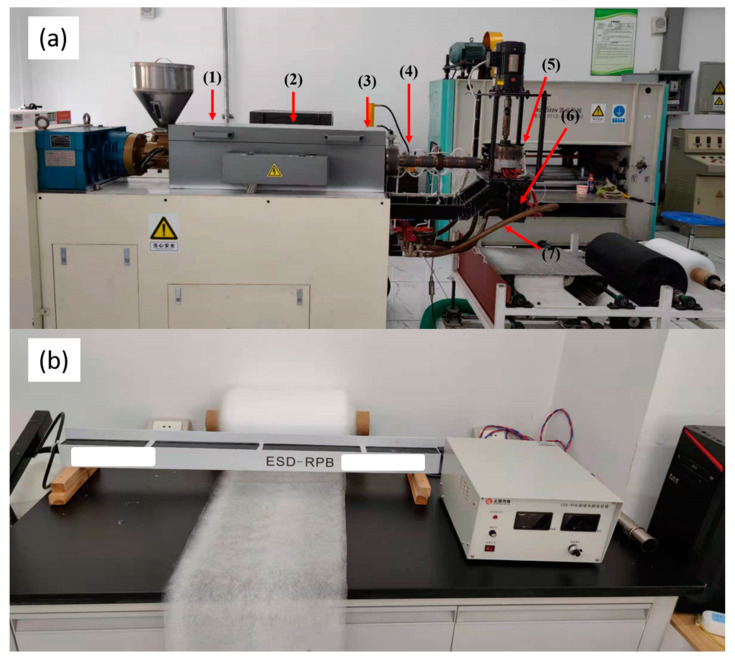
Images of (**a**) the melt blowing machine and (**b**) the electrostatic electret machine.

**Figure 2 polymers-13-03773-f002:**
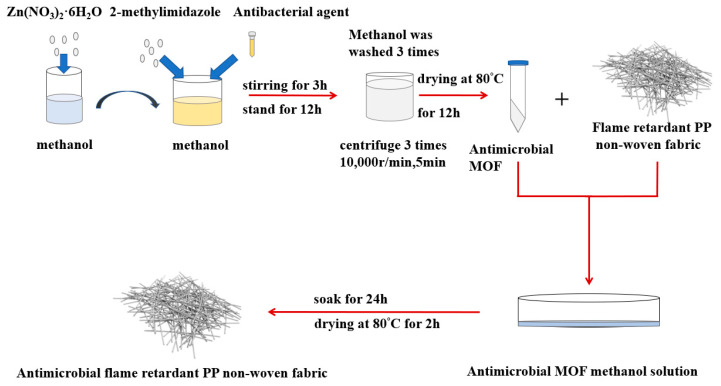
Flow chart of the preparation of polypropylene/zeolite imidazole framework-8 (PP/ZIF-8) melt-blown electrospun composite membranes.

**Figure 3 polymers-13-03773-f003:**
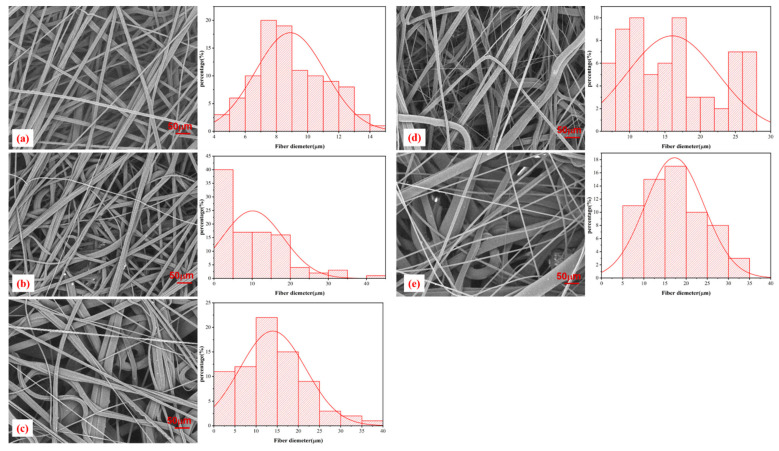
SEM images of melt-blown nonwoven fabrics: (**a**) PP, (**b**) PP-6, (**c**) PP-10, (**d**) PP-14, and (**e**) PP-18.

**Figure 4 polymers-13-03773-f004:**
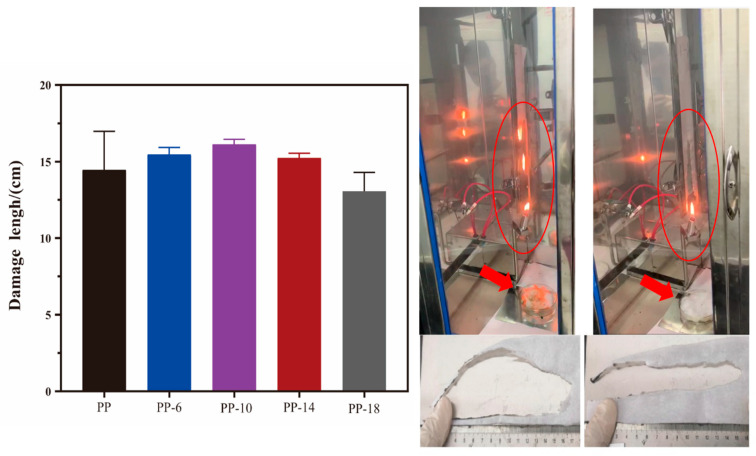
Diagram and images for vertical combustion damage results.

**Figure 5 polymers-13-03773-f005:**
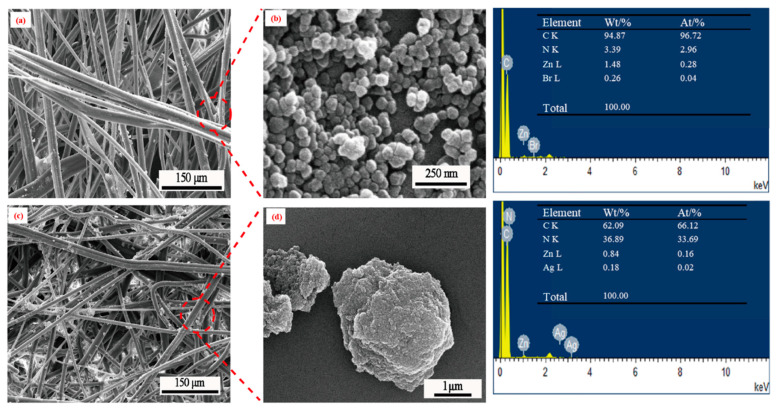
SEM and EDS analyses of ZIF-8@ melt-blown nonwoven fabrics and Ag@ZIF-8@ melt-blown nonwoven fabrics. (**a**) ZIF-8@melt-blown nonwoven fabrics (**c**) Ag@ZIF-8 melt blown nonwoven fabric (**b**,**d**) are the magnify of ZIF-8 and Ag@ZIF-8 nanoparticle on non-woven fabric, respectively3.4. FTIR Analysis.

**Figure 6 polymers-13-03773-f006:**
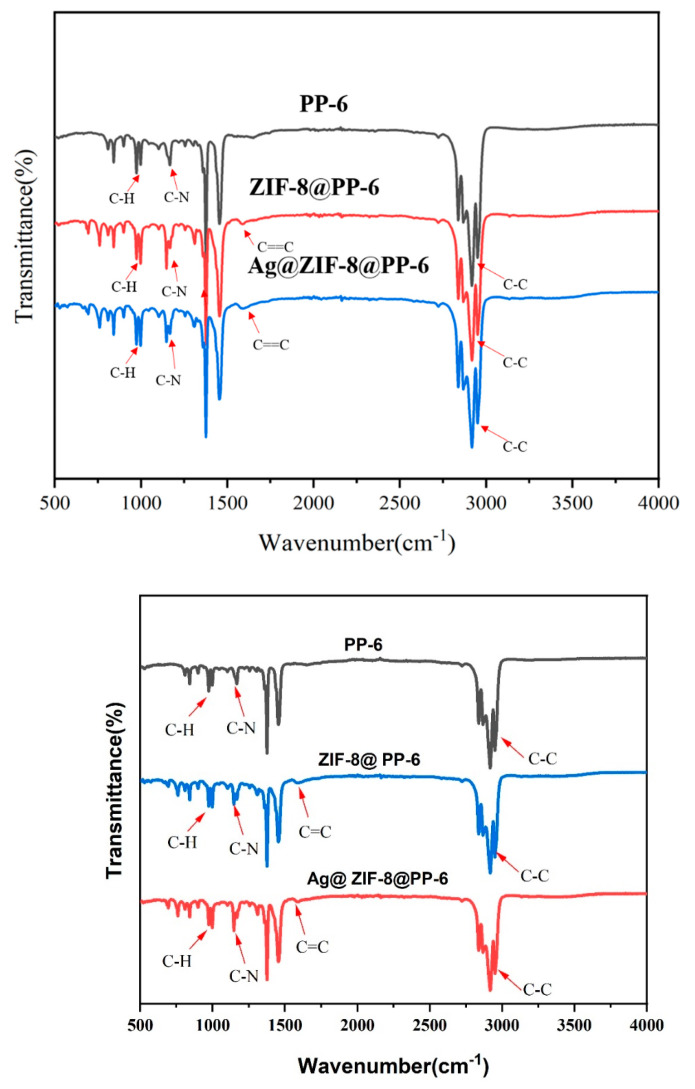
FTIR-ATR spectra of the grafted samples, including PP-6, ZIF-8@PP-6, and Ag@ZIF-8@PP-6.

**Figure 7 polymers-13-03773-f007:**
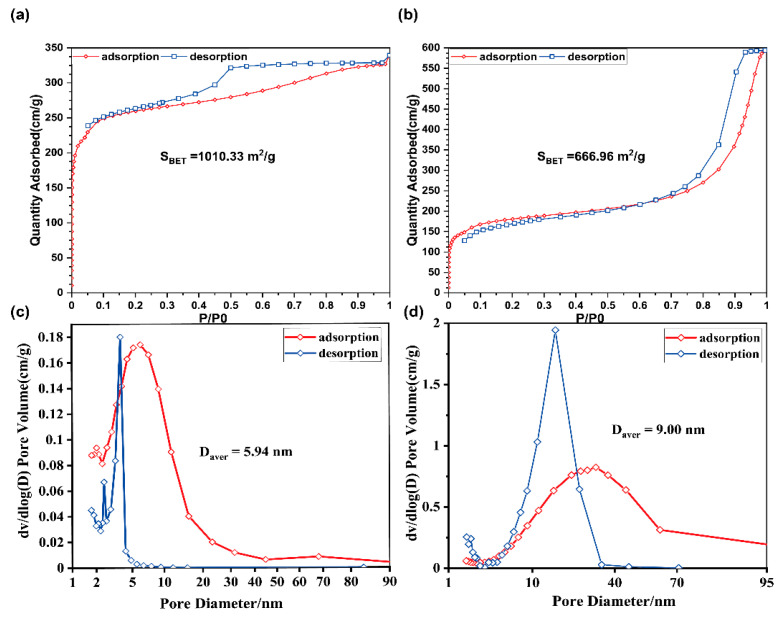
BET analysis and pore size of the as-prepared materials: the adsorption–desorption curve and pore size of (**a**/**c**) ZIF-8 and (**b**/**d**) Ag@ZIF-8 nanomaterials.

**Figure 8 polymers-13-03773-f008:**
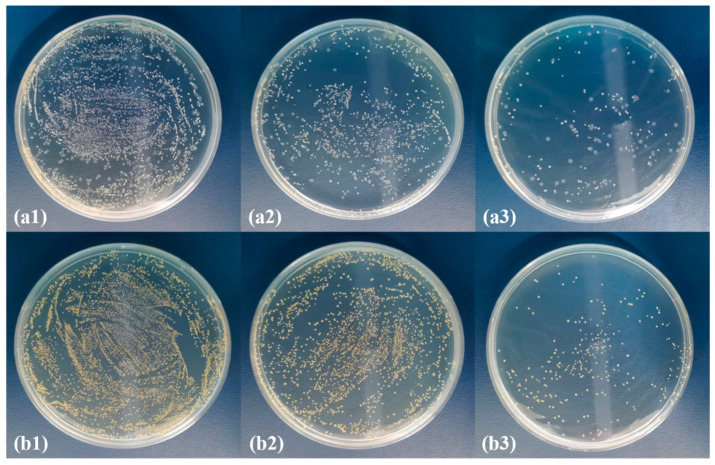
Antibacterial effect against (**a1**–**a3**) *Escherichia coli* and (**b1**–**b3**) *Staphylococcus aureus* of (**a1**,**b1**) the control group, (**a2**,**b2**) ZIF-8@ melt-blown nonwoven fabrics, and (**a3**,**b3**) Ag@ZIF-8@ melt-blown nonwoven fabrics.

**Figure 9 polymers-13-03773-f009:**
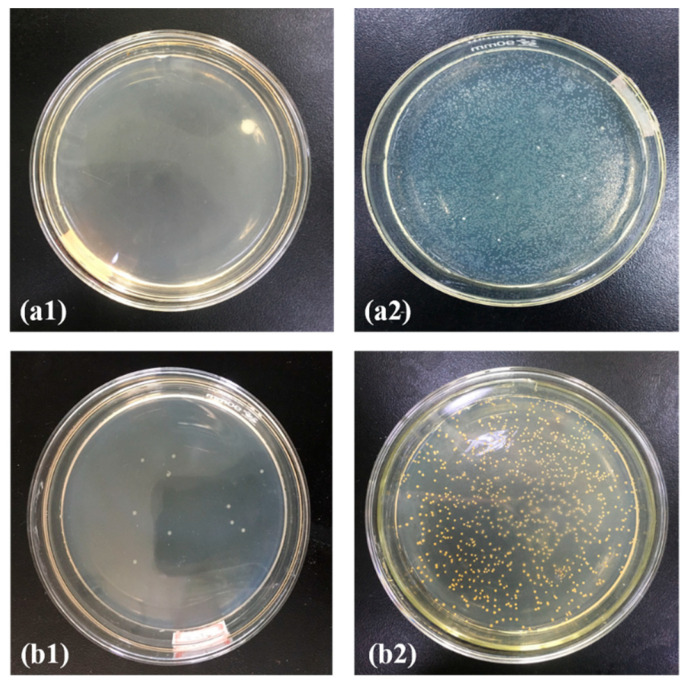
Ag@ZIF-8@ melt-blown nonwoven fabric electrostatic electret antibacterial test charts: (**a**) *E. coli* test chart, (**b**) Staphylococcus aureus test chart, 1. electrostatic electret treatment, and 2. non-electrostatic electret treatment.

**Figure 10 polymers-13-03773-f010:**
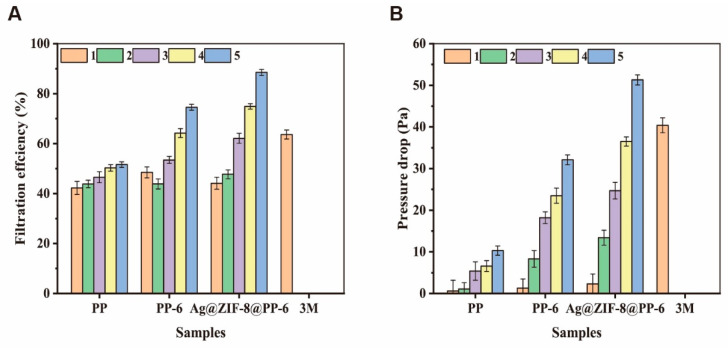
The test results of the filtration effect (**A**) and pressure drop (**B**) of various types of melt-blown nonwoven fabrics with different number of layers.

**Table 1 polymers-13-03773-t001:** Processing parameters.

Zone (1) °C	Zone (2) °C	Zone (3) °C	Pipeline (4) °C	Metering Pumps (5) °C	Nozzles Die Assembly Temperature (6) °C	Hot Air Outlet (7) °C
165	240	270	290	265	255	235

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
