# Peer review of "Preparation of Ag@ZIF-8@PP Melt-Blown Nonwoven Fabrics: Air Filter Efficacy and Antibacterial Effect"

_polymers, 2021, doi:10.3390/polym13213773_

Round 1
Reviewer 1 Report
- There are no keywords attached to the submitted manuscript.
- Check the number orders of the subtitles such as experimental titles with no number.
- There are some meaningless images in Figure 2 such as Flame retardant PP non-woven fabric, antimicrobial MOF, and antimicrobial flame retardant PP non-woven fabric.
- The introduction part should be enhanced with additional paragraphs.
- Figure 6 should be enhanced to be more readable.
Author Response
Thank you for your valuable comments. You have made changes and answers based on your questions. Please check the attachment.

Reviewer 2 Report
Major revision is needed. 1. Why add the flame retardant ?From figure 4 results, adding flame retardants makes no sense.
2. Please give detailed preparation parameters of part 2.2, 2.3, and 2.4 including concentration, dosage.
3. Is it the optimized preparation process of silver in Ag@ZIF-8@melt-blown nonwoven fabrics? The amount of silver is so low from your EDS analyses, so the antibacterial effects of Ag@ZIF-8@melt-blown nonwoven fabrics are not very well.
4. Line 331. The caption of figure 9 should be added.
5. What are the advantages over fabrics currently in use? 6. Manuscript writing is very poor, please revised this paper carefully.
Author Response

(The authors gave the same response as above.)

Round 2
Reviewer 2 Report
- The figure caption is very confused. Two Figure 6 are in this paper. Please revise including the text.
- Please pay attention to figure 6 (line 277). two figures are in Figure 6. The same case can be find in Figure 2 (line 166).
- Line 302. "pore diameter being 8.9960 nm" should be "pore diameter being 9.00 nm" based on the line 291 description. The same case can be find in Figure 6 c,d (line 306).
- "hours" and "h" are presented in manuscript, especially in Figure 2. Very confusing. Please unified.
- The bar is too small in Figure 5d than in a, b, and c.
- The text should be carefully revised including language and figure.
